# Immune Response following BNT162b2 mRNA COVID-19 Vaccination in Pediatric Cancer Patients

**DOI:** 10.3390/cancers15092562

**Published:** 2023-04-29

**Authors:** K. L. Juliëtte Schmidt, Noël M. M. Dautzenberg, Peter M. Hoogerbrugge, Caroline A. Lindemans, Stefan Nierkens, Gaby Smits, Rob S. Van Binnendijk, Louis J. Bont, Wim J. E. Tissing

**Affiliations:** 1Princess Máxima Center for Pediatric Oncology, Heidelberglaan 25, 3584 CS Utrecht, The Netherlands; 2Department of Pediatric Infectious Diseases and Immunology, Wilhelmina Children’s Hospital, University Medical Centre Utrecht, Lundlaan 6, 3584 EA Utrecht, The Netherlands; 3Center for Translational Immunology, University Medical Center Utrecht, Heidelberglaan 100, 3584 CX Utrecht, The Netherlands; 4Centre for Immunology of Infectious Diseases and Vaccines, National Institute for Public Health and the Environment, Antonie van Leeuwenhoeklaan 9, 3721 MA Bilthoven, The Netherlands; 5Department of Pediatric Oncology and Hematology, Beatrix Children’s Hospital, University of Groningen, University Medical Center Groningen, Hanzeplein 1, 9713 GZ Groningen, The Netherlands

**Keywords:** pediatrics, cancer, vaccination, SARS-CoV-2

## Abstract

**Simple Summary:**

Children with cancer experience a more severe SARS-CoV-2 infection and more often die due to COVID-19 than their healthy peers. Vaccination against SARS-CoV-2 is therefore recommend in children with cancer but little is known about their immune response following vaccination. With this study we aimed to investigate the effect of a 2- and/or 3-dose vaccination series in children who are undergoing active cancer treatment and in children that finished their cancer treatment. The results from this study show that compared to 2-dose vaccination, a 3-dose vaccination series was more effective in boosting antibody levels and is therefore of value for children undergoing active cancer treatment. The findings from this study provide evidence for the significance of COVID-19 vaccination in children undergoing cancer treatment and are also important for future vaccination strategies in children with cancer.

**Abstract:**

COVID-19 vaccinations are recommended for children with cancer but data on their vaccination response is scarce. This study assesses the antibody and T-cell response following a 2- or 3-dose vaccination with BNT162b2 mRNA COVID-19 vaccine in children (5–17 years) with cancer. For the antibody response, participants with a serum concentration of anti-SARS-CoV-2 spike 1 antibodies of >300 binding antibody units per milliliter were classified as good responders. For the T-cell response, categorization was based on spike S1 specific interferon-gamma release with good responders having >200 milli-international units per milliliter. The patients were categorized as being treated with chemo/immunotherapy for less than 6 weeks (Tx < 6 weeks) or more than 6 weeks (Tx > 6 weeks) before the first immunization event. In 46 patients given a 2-dose vaccination series, the percentage of good antibody and good T-cell responders was 39.3% and 73.7% in patients with Tx < 6 weeks and 94.4% and 100% in patients with Tx > 6 weeks, respectively. An additional 3rd vaccination in 16 patients with Tx < 6 weeks, increased the percentage of good antibody responders to 70% with no change in T-cell response. A 3-dose vaccination series effectively boosted antibody levels and is of value for patients undergoing active cancer treatment.

## 1. Introduction

Since the first description of a novel coronavirus in pneumonia patients identified in Wuhan, China, in December 2019 [1], Coronavirus disease 2019 (COVID-19) has rapidly spread with over 750 million cases worldwide and more than 6.8 million deaths due to COVID-19 [2]. Earlier research found adult cancer patients to be more vulnerable to SARS-CoV-2 as they were more often admitted to the ICU and had higher mortality rates compared to COVID-19 patients without cancer [3].

Although, compared to adults, SARS-CoV-2 infections in pediatric cancer patients seem to follow a relatively mild clinical course [4,5,6,7], pediatric cancer patients still have a higher morbidity and mortality than the general pediatric population. Recent data from the Global Registry of COVID-19 in Childhood Cancer (GRCCC) documented about 67.4% of the children being hospitalized, 19.9% having a severe or critical SARS-CoV-2 infection and ultimately 3.8% (high income country 1.3%) of the children overall dying due to COVID-19 [8]. In the general pediatric population, severe disease rates vary from 3 to 7% [9,10,11], with about 2–7% of the children being hospitalized [11,12] and mortality rates varying from 0.01% to 0.3% [9,10,11,12]. Besides the increased COVID-19-related morbidity and mortality in pediatric cancer patients, over 50% of the children had their anticancer treatments modified or delayed due to a COVID-19 infection [8], potentially compromising outcomes.

Because of their vulnerability, in 2021, pediatric cancer patients aged 5 years and older were given priority in the national COVID-19 vaccination program in the Netherlands where they were given a 2-dose series of the BNT162b2 (Pfizer/BioNTech) mRNA COVID-19 vaccine. Based on data from adult studies, patients 12 years and older were also offered an additional third vaccination [13]. However, data on the immune response in pediatric cancer patients following SARS-CoV-2 vaccination during or even after, intensive chemotherapy treatment is still largely lacking [14]. Moreover, the timing of the vaccination is also a topic of discussion for pediatric patients undergoing cancer treatment [15,16,17] and for patients who completed their treatment [16,17].

Therefore, in this study we aimed to assess the immune response after SARS-CoV-2 vaccination in children with cancer by investigating both the humoral and cellular response following a 2- or 3-dose series of the BNT162b2 mRNA COVID-19 vaccine, both during and after the end of treatment.

## 2. Materials and Methods

### 2.1. Study Design and Participants

This study was a prospective, observational, cohort study conducted at the Princess Máxima Center for Pediatric Oncology in Utrecht, the Netherlands.

We included patients aged 5–17 years, who were scheduled for a 2-dose series of 10 μg (5–11 years) or 30 μg (12–17 years) BNT162b2 (Pfizer/BioNTech) mRNA COVID-19 vaccine as part of the Dutch vaccination program. Patients treated at the Princess Máxima Center because of hematological, solid or neurological malignancies, or allogenic stem cell transplantation because of non-malignant disease, were identified from electronic medical records and asked to participate in blood sampling.

Written informed consent was obtained from all participants in this study. In the cases of patients between 12 and 15 years of age, both parents/legal guardians and patients needed to give informed consent in order to be able to participate. In the age category 5–11 years, both parents/legal guardians were asked for informed consent. This study was approved by the Medical Research Ethics Committee (MREC) Utrecht and Central Committee on Research involving Human Subjects (CCMO (NL78187.041.21, 21-430/D, EudraCT number: 2021-003388-90). It was conducted in accordance with the declaration of Helsinki and Good Clinical Practices guidelines.

### 2.2. Procedures/Blood Sample Collection and Laboratory Analyses

To assess the antibody and T-cell response, blood was sampled 28 (21–42) days after the 2nd vaccination. For immunocompromised children aged 12 and above, at a later stage, a 3rd vaccination dose was included in the national vaccination program. Patients (12–17 years) already included in the study, who received a 3rd vaccination, were asked for permission for an extra blood sample 28 (21–42) days after the 3rd vaccination.

Blood was drawn using an implanted port or venepuncture and collected in EDTA tubes, serum separator tubes, heparin sodium tubes and heparin lithium tubes.

### 2.3. Antibody Response

Total immunoglobulin G (IgG) antibody concentrations against SARS-CoV-2 spike 1 and nucleoprotein (N) were measured simultaneously using a bead-based assay, as previously described [18].

IgG concentrations were calibrated against the international standard for human anti-SARS-CoV-2 immunoglobulin (20/136 NIBSC standard) and reported as binding antibody units per mL (BAU/mL) [18]. Anti-S1 levels >10 BAU/mL were considered to be positive [19]. An anti-N level >14.3 BAU/mL was used as a criterion to distinguish past infection from COVID-19 vaccination as previously described [20], since anti-N antibodies are only found in cases of a clinical infection.

### 2.4. T-Cell Response

SARS-CoV-2-specific T-cell responses were measured using IFN-γ ELISA [SARS-CoV-2 interferon gamma release assay (EUROIMMUN, Lübeck, Germany)] according to the manufacturer’s instructions [21,22]. The amount of IFN-γ released from T cells was expressed in milli-international units per milliliter (mIU/mL). As recommend by the manufacturer, the following thresholds were used: non-responders <100 mIU/ml, low responders 100–200 mIU/mL, and good responders >200 mIU/mL [21,23]. To correct for background interferon gamma levels of each individual sample, the unstimulated control (CoV-2 IGRA BLANK) was subtracted from the SARS-CoV-2-stimulated tube (CoV-2 IGRA TUBE). SARS-CoV-2-stimulated tubes with concentration values above the highest calibration sample (>1890 mIU/mL) were corrected to this maximum value. Samples with negative values for mitogen stimulation (CoV-2 IGRA STIM) were excluded from analysis. The CoV-2 IGRA TUBE values were set to 0 in samples that had a negative CoV-2 IGRA BLANK and CoV-2 IGRA TUBE, but a positive CoV-2 IGRA STIM.

### 2.5. Data Collection

Demographic data, medical history, medication use and treatment status were collected using electronic medical records. The dates of the 1st, 2nd and, where applicable, 3rd vaccination were provided by each participant.

### 2.6. Outcomes

The primary study parameter was the SARS-CoV-2 spike 1-specific antibody concentration at 28 (21–42) days after the 2nd vaccination. We classified participants with anti-S1 levels >300 BAU/mL as responders, between 10 and 300 BAU/mL as low responders and <10 BAU/mL as non-responders, conforming with other studies [24,25].

The secondary endpoints were the SARS-CoV-2 spike 1-specific antibody concentration at 28 days after the 3rd vaccination and the SARS-CoV-2-specific T-cell responses.

### 2.7. Statistical Analysis

Based on their COVID-19 status, patients were divided into two groups: the Vaccination Only group consisting of patients without a history of SARS-CoV-2 infection and the Hybrid Immunity group consisting of patients with a positive test for SARS-CoV-2 before or after COVID-19 vaccination. In the Hybrid Immunity group, patients with the same number of immunizing events were grouped together where an immunizing event was defined as a COVID-19 vaccination or a SARS-CoV-2 infection. There had to be at least 10 days between a vaccination and an infection to be considered individual immunizing events. Patients with 2 immunizing events (1 vaccination + 1 infection) and patients with 3 immunizing events (2 vaccinations + 1 infection) were included in the analysis. No distinction was made between the sequence of vaccinations and infection. Patients with a SARS-CoV-2 infection before or after a 1-dose vaccination were included in the 2 immunizing events group, where patients with a SARS-CoV-2 infection before, after or between a 2-dose vaccination were included in the 3 immunizing events group. The 2 and 3 immunizing events patients were compared to COVID-19 naïve patients with a 2-dose or 3-dose vaccination series.

Based on treatment status, all participants were further divided into three groups consisting of patients who were treated with immuno- or chemotherapy <6 weeks before the first immunizing event (Tx < 6 weeks), >6 weeks before the first immunizing event (Tx > 6 weeks) or patients who never received chemo- or immunotherapy (No Tx). Immunotherapy was defined as treatment with monoclonal antibodies or CAR T-cell therapy. A 6 weeks cut-off point was used as most children who are undergoing active cancer treatment have an intense treatment regimen, receiving chemotherapy on a 3-week basis which is sometimes delayed due to toxicity. It was therefore assumed that all children who received chemotherapy in the last 6 weeks are children that are undergoing active cancer treatment.

Descriptive statistics were used to describe the different groups of participants. To describe continuous variables, the median and range were used. Antibody levels and the amount of IFN-γ released by SARS-CoV-2-specific T-cells were determined at the different time points in the Tx < 6 weeks, Tx > 6 weeks and No Tx groups and the Mann–Whitney U test was used to analyze differences between the Tx < 6 weeks and Tx > 6 weeks groups. For each treatment group, the percentage of good responders, low responders and non-responders was calculated. Statistical analyses were performed using IBM SPSS Statistics version 26.0.0.1 and GraphPad Prism version 9.3.0 for Windows (GraphPad Software, San Diego, CA, USA, www.graphpad.com, accessed on 27 January 2022).

## 3. Results

### 3.1. Patient Characteristics

Between the start of the study in July 2021 and May 2022, 86 children were enrolled. Seven children were excluded due to having only one immunizing event (*n* = 5), being infected more than once before vaccination (*n* = 1) or being treated with REGEN-COV (*n* = 1). Table 1 shows the baseline demographics and characteristics of the remaining 73 patients (6 patients not depicted due to an aberrant treatment group, as discussed later). Of the 73 patients, 28 (38.4%) children were aged 5–11 years and 45 (61.6%) children were aged 12–17 years; the median (range) age was 13 (5–17) years and 49.3% were male. Most children had a hematologic malignancy with acute lymphoblastic leukemia (ALL) as the most common diagnosis (38.4%). Ninety-two percent of the children had a treatment regimen that included chemotherapy. Seventeen children received a hematopoietic stem cell transplantation (HSCT) and/or CAR T-cell therapy (CAR T). In each of these children, there had been at least 3 months between their 1st vaccination and last HSCT/CAR T treatment.

In the Vaccination Only group, all children aged 5–11 years were vaccinated with a 2-dose series of the BNT162b2 vaccine while in the children aged 12–17 years, 16 children received a 2-dose and 21 children a 3-dose vaccination scheme with the BNT162b2 vaccine. As not all children gave permission for an extra blood sample after the 3rd vaccination, data about the antibody response after a 3-dose vaccination are only available for 16 of those 21 children. Appendix A shows the baseline demographics and characteristics for patients without a history of COVID-19 (Vaccination Only) and Appendix A shows the baseline demographics and characteristics for previously infected patients that experienced 2 or 3 immunizing events (Hybrid Immunity).

### 3.2. Humoral Immune Response

The anti-SARS-CoV-2 spike 1 (anti-S1) antibody response at 28 (21–42) days after 2-dose vaccination was available for 50 patients without a history of SARS-CoV-2 (Table 1). All four patients without a history of chemo- or immunotherapy were good responders (>300 BAU/mL) after a 2-dose vaccination (not included in further analyses) (Table 2).

One patient without a history of SARS-CoV-2 infection did not fulfill the criteria for Tx < 6 weeks (<6 weeks between chemotherapy and 1st vaccination) but did start chemotherapy for a newly discovered tumor between the 1st and 2nd vaccination. This patient (not included in further analyses) had an antibody concentration of 197 BAU/mL and a spike-specific T-cell response of 1024 mIU/mL after a 2-dose vaccination.

In the patients treated with chemo- or immunotherapy, median (range) antibody titers were significantly lower in the Tx < 6 weeks group compared to the Tx > 6 weeks group (91 BAU/mL [0.10–5091] vs. 5310 BAU/mL [56–17,061]; *p* < 0.0001; Figure 1A and Table 2). Of the patients with Tx < 6 weeks, 11 (39.3%) were good responders, 12 (42.9%) were low responders and 5 (17.9%) were non-responders. Of the patients with Tx > 6 weeks, all responded to vaccination: 17 (94.4%) were good responders and 1 (5.6%) patient was a low responder (Table 2). Figure 1B depicts the antibody concentration in relation to the number of days between the last chemo/immunotherapy treatment and 1st vaccination for the different diagnosis groups.

For 16 patients without a history of SARS-CoV-2 infection, data about the SARS-CoV-2-specific antibody concentration after a 3rd vaccination were available (Figure 1C, Table 2). Compared to 2-dose vaccinations, the median antibody (range) levels increased after a 3-dose vaccination to 546 (62–5645) BAU/mL in patients with Tx < 6 weeks and 20,573 (5404–35,596) BAU/mL in patients with Tx > 6 weeks. Three of the five patients in the Tx < 6 weeks group that were non-responders after a 2-dose vaccination showed seroconversion after an additional 3rd vaccination (the other 2 patients did not opt for a 3rd vaccination). In both the Tx < 6 weeks and Tx > 6 week groups, the percentage of good responders also increased to 70% and 100%, respectively (Table 2).

In total, 14 patients had a history of SARS-CoV-2 infection and a 1-dose vaccination (Hybrid Immunity, 2 immunizing events). The median (range) antibody titer in patients with Tx < 6 weeks was 209 (0.09–3129) BAU/mL (Table 2, Figure 1A) with 44.4% of these patients being good responders. All patients in the Tx > 6 weeks or No Tx group were good responders (Table 2) Those with a past infection and having received two doses of vaccine (3 immunizing events, *n* = 12) are also depicted in Table 2 and Figure 1C.

Overall in the 3 immunizing events group, five children were not included in the previous analyses due to their treatment course being different from the three treatment groups (Tx < 6 weeks, Tx > 6 weeks, no Tx). Three of those patients had a history of SARS-CoV-2 and did not receive any chemo- or immunotherapy treatment in the 6 weeks before their infection as they got infected before their cancer diagnosis. However, during the time of their 2-dose vaccination series, they were all undergoing treatment for cancer, receiving chemotherapy and/or immunotherapy. The other two patients were not included due to an unknown treatment status, as the exact timing of their SARS-CoV-2 infection was unknown. The anti-S1 antibody concentrations and (when available) spike-specific T-cell responses in those five patients were: 278 BAU/mL and 80 mIU/mL (patient 1), 11 BAU/mL (patient 2), 470 BAU/mL and 1890 mIU/mL (patient 3), 6380 BAU/mL (patient 4), and 10,088 BAU/mL and 581 mIU/mL (patient 5).

### 3.3. Cellular Immune Response

For 34 patients without a history of SARS-CoV-2 infection, data were available about their T-cell response following a 2-dose vaccination. All patients (*n* = 3) without a history of chemotherapy or immunotherapy were good responders after two vaccinations. The median (range) amount of IFN-γ release was significantly lower in the Tx < 6 weeks group compared with the Tx > 6 weeks group (841 [0–1890] mIU/mL vs. 1890 [264–1890] mIU/mL; *p* = 0.0083). In the Tx > 6 weeks group, all but one patient reached the maximum value of 1890 mIU/mL. However, this one patient was still a good responder (Table 3, Figure 2A). This patient is not the same person as the low antibody responder that was previously described in the Tx > 6 weeks group. In the Tx < 6 weeks group, 14 (73.7%) patients were good responders, 1 (5.3%) patient was a low responder and 4 (21.1%) patients were non-responders (Table 3).

Figure 2B shows the relationship between T-cell response and number of days between last therapy and 1st vaccination for the different diagnosis groups in patients without a history of SARS-CoV-2 infection.

For 7 of the 34 patients without a history of SARS-CoV-infection, data were available about their T-cell response following an additional 3rd vaccination. The median (range) SARS-CoV-2-specific IFN-γ-release in the Tx < 6 weeks group was 669 (54–1890) mIU/mL after three vaccinations with one patient being a non-responder and the other two patients being good responders. All patients in the Tx > 6 weeks group were good responders (Table 3, Appendix A).

Of the 14 patients in the Hybrid Immunity group who were previously infected with SARS-CoV-2 and who received a 1-dose vaccination (2 immunizing events), 6 patients had data available about their T-cell response. In the Tx < 6 weeks group, one patient was a non-responder and two patients were good responders. In the Tx > 6 weeks group, one patient was a low responder and two patients were good responders (Table 3).

The median SARS-CoV-2-specific IFN-γ release in the Hybrid Immunity group after 3 immunizing events (SARS-CoV-2 infection + 2 vaccinations) is also depicted in Table 3 (and Appendix A). All patients in the Tx < 6 weeks and Tx > 6 weeks group were good responders.

Of all the patients without a history of SARS-CoV-2 in the Tx < 6 weeks group for whom both T-cell and antibody data were available, six (31.6%) had an adequate antibody and T-cell response after 2-dose vaccination and five (26.3%) had an inadequate antibody response and inadequate T-cell response. Eight (42.1%) children in the Tx < 6 weeks group had a good T-cell response but a non- or low antibody response (Table 4). There were no children with a good antibody response but a non- or low T-cell response.

## 4. Discussion

This study characterized the induction of hybrid immunity in children with cancer. Our results show that, despite being on active treatment, 39% of the pediatric patients without a history of SARS-CoV-2 infection had sufficient anti-SARS-CoV-2 antibodies and 74% had a sufficient spike-specific T-cell response after a 2-dose vaccination series using the BNT162b2 mRNA COVID-19 vaccine. This is in contrast to the immune responses following a 2-dose series in the Tx > 6 weeks group, where all patients had an adequate T-cell response and all but one patient had optimal antibody levels. A 3-dose vaccination series was shown to be effective in increasing the humoral immune response, and is therefore of value for patients undergoing active treatment.

Our findings of the added benefit of a 3rd vaccination are in line with other smaller studies in pediatric cancer patients [26,27] and adult cancer patients [28]. Poparn et al. measured an overall seroprotective rate of 67.4% in 43 adolescents aged 12–18 years after a 2-dose vaccination with significant differences between the on- and off-therapy groups. In the on-therapy group, 33.3% reached seroprotective levels which is comparable to the 39.3% as found in our study. All off-therapy group patients in the study of Poparn et al. had an adequate antibody response while we had one patient in the after active treatment group that was a low responder [26]. This patient with a history of osteosarcoma had to stop cancer treatment in the past due to unexplained severe myelosuppression that persisted for multiple months following chemotherapy. Although at the time of 1st vaccination the number of leukocytes was sufficient, this might still have a lingering effect on vaccination efficiency. This patient had an adequate T-cell response (1890 mIU/mL). Lehrnbecher et al. measured antibodies against the receptor-binding domain (RBD) of SARS-CoV-2 in 21 adolescent patients undergoing active cancer treatment using a cut-off value of 7 BAU/mL for a positive test result. They found detectable antibody levels in 76.2% of their patients after two vaccine doses which increased significantly to 90% after the 3rd vaccine dose [27]. We too found an increase in median antibody levels in the Tx < 6 weeks group after the 3rd dose with 70% of the children in this group becoming good responders. Furthermore all three children in the Tx < 6 weeks group that provided data on both the 2nd and 3rd vaccination, showed seroconversion after this additional 3rd vaccination. Likewise, Poparn et al. detected seroprotective levels (defined as a SARS-CoV-2 surrogate virus neutralization test [sVNT] >68%) in 27.3% of the adolescents after a 3-dose vaccination who were previously seronegative [26]. Although different methods and definitions of a sufficient antibody response were used in both the studies from Lehrnbecher et al. and Poparn et al. compared to our study, all three studies clearly showed that a 3rd vaccination increases antibody responses in patients undergoing active treatment [26,27]. Our data also described the antibody response in previously SARS-CoV-2 infected patients. These Hybrid Immunity patients with 2 immunizing events (1 vaccination and 1 episode of infection) had comparable or even higher antibody levels than 2-dose vaccination patients without a history of SARS-CoV-2. Forty-four percent of the patients in the Tx < 6 weeks group were good antibody responders after 2 immunizing events illustrating that the combination of SARS-CoV-2 infection and vaccination effectively boost immunity. This adequate immune response after a breakthrough infection or after a vaccination following infection has been described before in adults by Bates et al. A significantly larger boost in antibody response was seen in patients with a history of SARS-CoV-2 infection and 1- or 2-dose vaccination, compared to 2-dose vaccination, COVID-19 naïve patients [29].

An adequate cellular immune response was found in 73.7% of the COVID-19 naïve children undergoing active treatment after 2-dose vaccination. This is comparable to the results from Lehrnbecher et al. who found SARS-CoV-2-specific T-cells in 67% of their pediatric cancer patients [27]. This is also in line with a study in adult cancer patients where the percentages of spike-specific T-cell responses ranged from 53 to 67% [24]. In the Hybrid Immunity group the percentage of good T-cell responders increased from 66.7% after a 2-dose vaccination to 100% after an additional vaccination.

Please note that 42.1% of the patients without a SARS-CoV-2 infection in the Tx < 6 weeks group were non- or low antibody responders but good T-cell responders. This discordancy between humoral and cellular immune responses that we observed has been described before in adult cancer patients [24].

This study assessed the humoral and cellular immune response following COVID-19 vaccination but data about the clinical efficacy, that is the number of infections following vaccination is not yet completely available. Future follow up 1 year after the 2nd vaccination will reveal the long-term humoral and cellular immunity following vaccination and the clinical relevance of vaccination against SARS-CoV-2. However, although the clinical relevance is yet unknown, 73.7% of the children demonstrating an adequate T-cell response is still a promising result as new research shows the cellular immune response to play an important role in eliciting and maintaining an adequate immune response after infection [30] or vaccination [31]. Recently, it was shown that memory T-cells were able to cross-recognize different variants of the SARS-CoV-2 virus and were able to maintain this response during a longer period of time (at least 6 months) while the antibody response and memory B cells decreased during this time [31]. Several studies reported a waning humoral immune response following COVID-19 vaccination [32,33] or SARS-CoV-2 infection [34]. A substantial decrease in IgG levels and neutralizing antibody titers was seen in the 6 months following BNT162b2 COVID-19 vaccinations in adults [32]. This waning of immunity, typically starts 3–24 weeks after a full vaccination course [33]. Waning is even more rapid in immunocompromised patients and is also seen after a 3rd vaccination [35]. Additionally, in children with a history of SARS-CoV-2 infection, waning of the humoral immune response was seen in the 14 months following infection [34]. Therefore, future research is needed to optimize the booster schedule in children with cancer [36].

This study has several limitations. Although it had a larger sample size than previously published studies and included both antibody and T-cell responses, statistical analyses were still limited by the small sample size and heterogeneous population. Further research is needed to identify risk factors for impaired humoral and cellular immune response following vaccination. Mavinkurve-Groothuis et al. showed that the absolute lymphocyte count prior to vaccination predicted the vaccination response to H1N1 influenza virus vaccination in pediatric cancer patients. This was demonstrated for different subpopulations of lymphocytes; no protective vaccination response could be found in patients with CD4+ T cell counts less than 200/mm^3^ [37]. Furthermore our study did not have enough power to compare differences in immune response between different diagnosis groups. Research among adult patients with solid tumors found most of them to have an adequate antibody response and that percentages of suboptimal or non-responders ranged from 7 to 16% in the different chemotherapy and immunotherapy groups [24]. A recent meta-analysis showed a significant difference in seroconversion rates between adult patients with solid tumors and adults patients with hematological malignancies (94% vs. 65%, *p* < 0.001) [38]. Lehrnbecher et al. also described that lower antibody levels were found in adolescents with hematological malignancies compared to solid tumors [27]. Our results show this same trend of a lower immune response in patients with hematological malignancies vs. solid tumors, with roughly 40% of the children that were being actively treated for ALL were good responders while in children being actively treated for a solid or CNS tumor this percentage was about 50% (Figure 1B). Next to tumor type, the treatment regimen may also influence immune responses as adolescent patients with more intense treatment regimens had lower titers compared to patients in maintenance therapy [27].

Despite these limitations, the heterogeneity of the included patients is also one of the strengths of this study. Both the studies from Lehrnbecher et al. and Poparn et al. reported data from children older than the age of 12 and all without a history of SARS-CoV-2 infection [26,27]. The data on the immune response following COVID-19 vaccination are therefore scarce in children aged 5–11 years and in children with a hybrid immunity. As this study included children with these characteristics, the current results contribute to our knowledge on COVID-19 vaccination during and after cancer treatment and also specifically in younger children and in previously infected children.

## 5. Conclusions

Overall, this study provides insight into the immune responses following vaccination and SARS-CoV-2 infection in children with cancer. Based on these results, we recommend that, given the high percentage (70%) of good antibody responders in the Tx < 6 weeks group after a 3-dose series of the BNT162b2 mRNA COVID-19 vaccine, children undergoing active cancer treatment should be eligible for a three-dose vaccination regimen in order to protect them from COVID-19 infections.

## Figures and Tables

**Figure 1 cancers-15-02562-f001:**
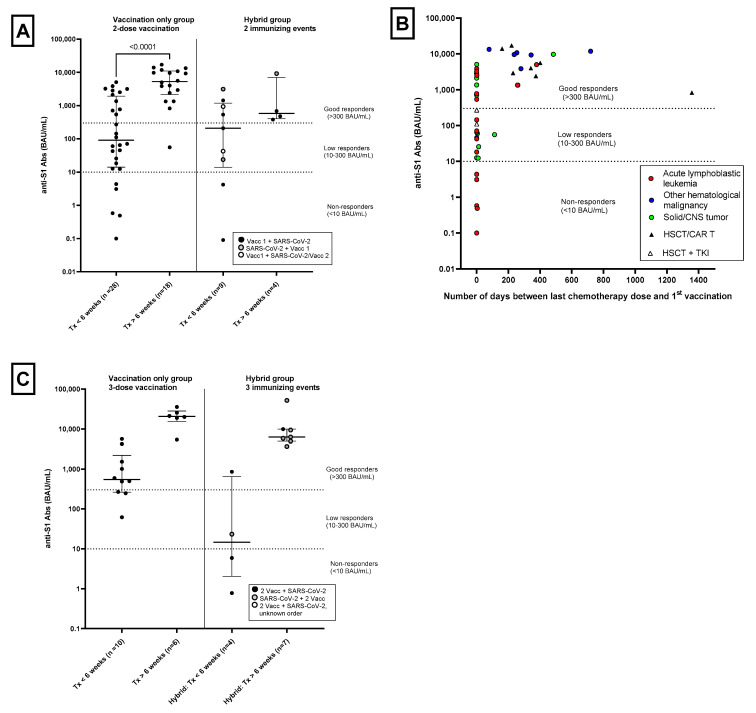
SARS-CoV-2-specific antibody (anti-S1 Abs) levels in patients aged 5–17, vaccinated with a 2- or 3-dose series of BNT162b2 mRNA COVID-19 vaccine or having 2 or 3 immunizing events. The horizontal dashed lines indicates thresholds for: non-responders (<10 BAU/mL), low responders (10–300 BAU/mL) or good responders (>300 BAU/mL). BAU/mL = binding antibody units per milliliter. (**A**) Anti-S1 antibody levels at 4 weeks after 2-dose vaccination or 2 immunizing events in all patients aged 5–17. Patients were divided into two groups consisting of patients with the last immuno- or chemotherapy dose <6 weeks before 1st vaccination or 1st immunizing event (Tx < 6 weeks) or >6 weeks before 1st vaccination or 1st immunizing event (Tx > 6 weeks). Vaccination Only group: *p* < 0.001 as determined by the Mann–Whitney U test. In the Hybrid Immunity group, patients were further categorized as having 1 vaccination followed by SARS-CoV-2 infection (black circles), being previously infected with SARS-CoV-2 followed by 1 vaccination dose (grey circles) or 1 vaccination followed by another vaccination and infection that occurred <10 days of each other counting as 1 immunizing event (open circles). (**B**) Anti-S1 antibody levels at 4 weeks after 2-dose vaccination in all patients aged 5–17 years without a history of SARS-CoV-2 infection and the number of days between last immuno- or chemotherapy dose and 1st vaccination. Red circles indicate patients with acute lymphoblastic leukemia; blue circles, patients with other hematological malignancies; green circles, patients with a solid or central nervous system tumor; black triangles, patients with a history of HSCT and/or CAR T-cell therapy; white triangles, patients with a history of HSCT who were being treated with a tyrosine kinase inhibitor (TKI). (**C**) Anti-S1 antibody levels at 4 weeks after 3-dose vaccination in patients aged 12–17 years or 3 immunizing events in patients aged 5–17 years. Patients were divided into two groups consisting of patients with the last immuno- or chemotherapy dose < 6 weeks before 1st vaccination or 1st immunizing event (Tx < 6 weeks) or >6 weeks before 1st vaccination or 1st immunizing event (Tx > 6 weeks). In the Hybrid Immunity group, patients were further categorized as having 2 vaccinations followed by SARS-CoV-2 infection (black circles), being previously infected with SARS-CoV-2 followed by 2-dose vaccination (grey circles) or an unknown order of 2-dose vaccination + SARS-CoV-2 infection (open circles).

**Figure 2 cancers-15-02562-f002:**
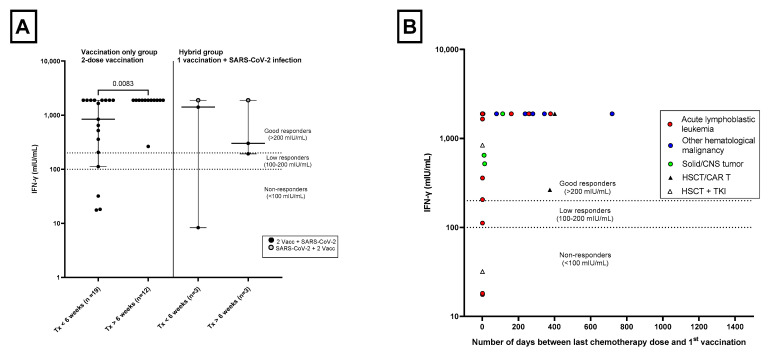
The amount of interferon-gamma (IFN-γ) released by SARS-CoV-2-specific T-cells in patients aged 5–17, 4 weeks after a 2-dose series of BNT162b2 mRNA COVID-19 vaccine or 2 immunizing events. The horizontal dashed lines indicates: non-responders (<100 mIU/mL), low responders (100–200 mIU/mL) or good responders (>200 mIU/mL). mIU/mL= milli-international units per milliliter. (**A**) SARS-CoV-2-specific T-cell response in the Vaccination Only group and the Hybrid group (1 vaccination followed by a SARS-CoV-2 infection). Patients were divided into two groups consisting of patients with the last dose of immuno- or chemotherapy <6 weeks before 1st vaccination or 1st immunizing event (Tx < 6 weeks) or >6 weeks before 1st vaccination or 1st immunizing event (Tx > 6 weeks). Vaccination Only group: *p* = 0.0083 as determined by Mann–Whitney U test. One patient in the Vaccination Only Tx < 6 weeks group with a value of 0 mIU/mL is not depicted. (**B**) SARS-CoV-2-specific T-cell response and the number of days between last immuno- or chemotherapy dose and 1st vaccination in patients aged 5–17 years without a history of SARS-CoV-2 infection. Red circles indicate patients with acute lymphoblastic leukemia; blue circles, patients with other hematological malignancies; green circles, patients with a solid tumor or central nervous system tumor; black triangles, patients with a history of HSCT and/or CAR T-cell therapy; white triangles, patients with a history of HSCT who were being treated with a tyrosine kinase inhibitor (TKI). One patient with a value of 0 mIU/mL and 1 day between his last chemotherapy and first vaccination is not depicted.

**Table 1 cancers-15-02562-t001:** Baseline demographics and patient characteristics (*n* = 73).

	Tx < 6 Weeks ^a^	Tx > 6 Weeks ^a^	No Tx ^a^
	*n* = 39 (%)	*n* = 28 (%)	*n* = 6 (%)
Per age category, *n* (%)			
5–11 years	21 (53.8)	7 (25)	0
12–17 years	18 (46.2)	21 (75)	6 (100)
Age, years (median, range)	11 (5–17)	14 (5–17)	14.5 (12–16)
Sex, *n* (%)			
Male	21 (53.8)	12 (42.9)	3 (50)
Female	18 (46.2)	16 (57.1)	3 (50)
Diagnosis, *n* (%)			
Acute lymphoblastic leukemia	20 (51.3)	8 (28.6)	0
Other hematological malignancies	6 (15.4)	11 (39.3)	4 (66.7)
Solid tumors	7 (17.9)	7 (25)	2 (33.3)
CNS tumors	5 (12.8)	0	0
Non-malignancies ^b^	1 (2.6)	2 (7.1)	0
History of chemotherapy, *n* (%)	39 (100)	28 (100)	0
History of immunotherapy, *n* (%)	5 (12.8)	7 (25)	0
Bevacizumab	3	0	
Blinatumomab	0	2	
Daratumumab	0	1	
Dinutuximab	1	0	
Rituximab	0	2	
Tocilizumab	0	1	
Tocilizumab, rituximab	1	0	
Basiliximab, infliximab, vedolizumab	0	1	
History of recent IVIG ^c^, *n* (%)	1 (2.6)	2 (5.3)	0
History of HSCT/CAR T ^d^, *n* (%)	3 (7.7)	14 (50)	0
Allo-SCT	2 (66.7)	7 (50)	
Auto-SCT	0	3 (21.4)	
CAR T	0	2 (14.3)	
CAR T + allo-SCT	0	2 (14.3)	
CAR T + auto-SCT	1 (33.3)	0	
Antibody response—number of patients with data available			
Vaccination group ^e^: 28 days after the 2nd vaccination	28	18	4
Hybrid group ^f^: 28 days after the 2nd immunizing event ^g^	9	4	1
Vaccination group: 28 days after the 3rd vaccination	10	6	0
Hybrid group: 28 days after the 3rd immunizing event	4	7	1
T-cell response—number of patients with data available			
Vaccination group: 28 days after the 2nd vaccination	19	12	3
Hybrid group: 28 days after the 2nd immunizing event	3	3	0
Vaccination group: 28 days after the 3rd vaccination	3	4	0
Hybrid group: 29 days after the 3rd immunizing event	4	5	0

^a^ Defined as receiving chemotherapy or immunotherapy less (Tx < 6 weeks) or more (Tx > 6 weeks) than 6 weeks before 1st vaccination or having no history of chemotherapy or immunotherapy (No Tx). ^b^ Non-malignancies: familial autosomal dominant primary immunodeficiency (RAC 2 mutation) (*n* = 1) and Fanconi anemia (*n* = 2). ^c^ IVIG (intravenous immunoglobulin) <2 months prior to 1st vaccination or SARS-CoV-2 infection. ^d^ Number of days between HSCT and 1st vaccination varied between 95 and 1925 days. ^e^ Vaccination Only group consisting of patients without a history of SARS-CoV-2 infection. ^f^ Hybrid Immunity group consisting of patients with a positive test for SARS-CoV-2 before or after COVID-19 vaccination. ^g^ An immunizing event was defined as a COVID-19 vaccination or a SARS-CoV-2 infection.

**Table 2 cancers-15-02562-t002:** Anti-SARS-CoV-2 spike 1 antibody response ^a^ following vaccination and/or SARS-CoV-2 infection.

	Treatment Group		
	Tx < 6 Weeks	Tx > 6 Weeks	No Tx
2-dose vaccination			
Number of patients (N)	28	18	4
Median (range) BAU/mL	91 (0.10–5091)	5310 (56–17,061)	2907 (1447–6837)
Non-responders	5 (17.9%)	0	0
Low responders	12 (42.9%)	1 (5.6%)	0
Good responders	11 (39.3%)	17 (94.4%)	4 (100%)
Hybrid (1 vaccination + SARS-CoV-2 infection) ^b^			
Number of patients (N)	9	4	1
Median (range) BAU/mL	209 (0.09–3129)	583 (379–9213)	6142
Non-responders	2 (22.2%)	0	0
Low responders	3 (33.3%)	0	0
Good responders	4 (44.4%)	4 (100%)	1 (100%)
3-dose vaccination			
Number of patients (N)	10	6	0
Median (range) BAU/mL	546 (62–5645)	20,573 (5404–35,596)	
Non-responders	0	0	
Low responders	3 (30%)	0	
Good responders	7 (70%)	6 (100%)	
Hybrid (2 vaccinations + SARS-CoV-2 infection) ^c^			
Number of patients (N)	4	7	1
Median (range) BAU/mL	14.6 (0.78–855)	6296 (3669–51,931)	11,540
Non-responders	2 (50%)	0	0
Low responders	1 (25%)	0	0
Good responders	1 (25%)	7 (100%)	1 (100%)

^a^ Antibody response group; non-responders <10 BAU/mL, low responders 10–300 BAU/mL, good responders >300 BAU/mL. ^b^ Patients with 2 immunizing events: either an infection followed by a vaccination, or an infection after a vaccination, or a vaccination followed by a vaccination and infection that occurred <10 days of each other counting as 1 immunizing event. ^c^ Patients with 3 immunizing events: infected before, between or after 2-dose vaccination.

**Table 3 cancers-15-02562-t003:** SARS-CoV-2-specific T-cell response ^a^ following vaccination and/or SARS-CoV-2 infection.

	Treatment Group		
	Tx < 6 Weeks	Tx > 6 Weeks	No Tx
2-dose vaccination			
Number of patients (N)	19	12	3
Median (range) mIU/mL	841 (0–1890)	1890 (264–1890)	1890 (1890–1890)
Non-responders	4 (21.1%)	0	0
Low responders	1 (5.3%)	0	0
Good responders	14 (73.7%)	12 (100%)	3 (100%)
Hybrid (1 vaccination + SARS-CoV-2 infection) ^b^			
Number of patients (N)	3	3	0
Median (range) mIU/mL	1408 (8.32–1890)	301 (193–1890)	
Non-responders	1 (33.3%)	0	
Low responders	0	1 (33.3%)	
Good responders	2 (66.7%)	2 (66.7%)	
3-dose vaccination			
Number of patients (N)	3	4	0
Median (range) mIU/mL	669 (54–1890)	1890 (935–1890)	
Non-responders	1 (33.3%)	0	
Low responders	0	0	
ood responders	2 (66.7%)	4 (100%)	
Hybrid (2 vaccinations + SARS-CoV-2 infection) ^c^			
Number of patients (N)	4	5	0
Median (range) mIU/mL	1498 (777–1890)	1890 (945–1890)	
Non-responders	0	0	
Low responders	0	0	
Good responders	4 (100%)	5 (100%)	

^a^ T-cell response group: non-responders <100 mIU/mL, low responders 100–200 mIU/mL, good responders >200 mIU/mL. ^b^ Patients with 2 immunizing events: either an infection followed by a vaccination or an infection after a vaccination or a vaccination followed by a vaccination and infection that occurred <10 days of each other counting as 2 immunizing events. ^c^ Patients with 3 immunizing events: infected before, between or after 2-dose vaccination.

**Table 4 cancers-15-02562-t004:** Antibody response ^a^ and T-cell response ^b^ group following 2-dose vaccination in patients without a history of SARS-CoV-2 infection.

	Tx < 6 Weeks	Tx > 6 Weeks	Total
Good antibody responder + good T-cell responder, *n* (%)	6 (31.6)	11 (91.7)	17 (54.8)
Non- or low antibody responder + good T-cell responder, *n* (%)	8 (42.1)	1 (8.3)	9 (29)
Non- or low antibody responder + non- or low T-cell responder, *n* (%)	5 (26.3)	0	5 (16.1)

**^a^** Antibody response group: non responders <10 BAU/mL, low responders 10–300 BAU/mL, good responders >300 BAU/mL. **^b^** T-cell response group: non responders <100 mIU/mL, low responders 100–200 mIU/mL, good responders >200 mIU/mL.

## Data Availability

The data that support the findings of this study are not publicly available but are available from the authors upon reasonable request.

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
