# Peer review of "Immune Response following BNT162b2 mRNA COVID-19 Vaccination in Pediatric Cancer Patients"

_cancers, 2023, doi:10.3390/cancers15092562_

Round 1

Reviewer 1 Report

The article is of high interest, well-written, with clear results. I think it should be published. However, I have some comments/questions for the authors

1) The scale used in Figures 1B and 2B may let the reader think that the vaccination started on the first day of chemotherapy in a large proportion of the patients. The figures may be improved by removing extreme values (see below) and/or cutting the X-axis to increase the scale in the left part.

2) The authors chose a 6-weeks cut-off from receiving or not a treatment. It appears reasonable but the authors should justify their choice and indicate somewhere why they took this decision.

3) It may be discutable to have included patients who received a treatment more than one year before the vaccination, especially for patients with no HSCT/CAR-T history, since we can anticipate that the response to vaccination may be good. The authors should at least discuss this point. Removing some patients from the analysis may help to improve the clarity of the figures (see above)

4) The authors claim in their discussion and their conclusion that this is the "first study to characterize...", or that this study "provides considerable insight" etc. I think their is no need to use such terms. The interest and the quality of the work is obvious for the reader, and the use of such formulations may appear pretentious.

Some typing errors were found within the text (not exhaustive):

-line 188 repetition "in in"

-line 374 SARS-COV-2 instead of Sars-CoV-2

Author Response

Response to Reviewer 1

The article is of high interest, well-written, with clear results. I think it should be published.

Response:

Thank you for the insightful and helpful comments. We hope that we have sufficiently addressed all comments.

However, I have some comments/questions for the authors

1) The scale used in Figures 1B and 2B may let the reader think that the vaccination started on the first day of chemotherapy in a large proportion of the patients. The figures may be improved by removing extreme values (see below) and/or cutting the X-axis to increase the scale in the left part.

Response 1:

Thank you for the suggestions. Most patients got vaccinated during their anti-cancer treatment in the first two weeks after their last chemotherapy as most patients have a 3-weekly chemotherapy regimen. We tried to improve the figure by cutting the X-axis and by removing extreme values but unfortunately this didn’t improve the figure. We however modified the titles of the X-axes of both figure 1B and figure 2B, changing the words “last chemotherapy” to “last chemotherapy dose”. We also made the same changes in the description of figure 1 and figure 2.

In the description of figure 1A (lines 294 & 295) “with immuno- or chemotherapy < 6 weeks” was changed to “with the last immuno- or chemotherapy dose < 6 weeks”.

In the description of figure 1B (line 300) “of days between last immuno- or chemotherapy and” was changed to “of days between last immuno- or chemotherapy dose and”.

In the description of figure 1C (line 307) “with immuno- or chemotherapy < 6 weeks” was changed to “with the last immuno- or chemotherapy dose < 6 weeks”.

In the description of figure 2A (line 365) “with immuno- or chemotherapy < 6 weeks” was changed to “with the last dose of immuno- or chemotherapy < 6 weeks”.

In the description of figure 2B ( line 368) “days between last immuno- or chemotherapy and” was changed to “days between last immuno- or chemotherapy dose and”.

In this way we hope to have made it more explicit that the indicated timespan is the number of days between the last dosage of chemotherapy and the first vaccination for a patient. We hope these modifications increase the readability of figure 1B and 2B.

2) The authors chose a 6-weeks cut-off from receiving or not a treatment. It appears reasonable but the authors should justify their choice and indicate somewhere why they took this decision.

Response 2:

Thank you for pointing this out. Most children in the study receive 3-weekly chemotherapy. However this is sometimes delayed due to toxicity. Based on this we made the assumption that children who received chemotherapy in the last six weeks, are children that are under active cancer treatment.  We agree with the comment that we should further explain why we have chosen this 6 weeks cut-off point and have therefore included the following sentences to the method section that mentions the categorization based on treatment status. 

Lines (198-202) “A 6 weeks cut-off point was used as most children who are under active cancer treatment have an intense treatment regimen, receiving chemotherapy on a 3-weekly basis which is sometimes delayed due to toxicity. It was therefore assumed that all children who received chemotherapy in the last 6 weeks, are children that are under active cancer treatment.”

3) It may be discutable to have included patients who received a treatment more than one year before the vaccination, especially for patients with no HSCT/CAR-T history, since we can anticipate that the response to vaccination may be good. The authors should at least discuss this point. Removing some patients from the analysis may help to improve the clarity of the figures (see above)

Response 3:

We appreciate your feedback. This study mainly intended to investigate the immune response during and after cancer treatment and gain more knowledge on this topic. We included figure 1B and 2B to give readers a short overview of possible differences between patients with different tumor types and possible differences between patients with and without a history of HSCT/ CAR T. Indeed we anticipated patients more than one year after stop of treatment to have a good response. However, since we have the data we think we have to present them to be complete.   

4) The authors claim in their discussion and their conclusion that this is the "first study to characterize...", or that this study "provides considerable insight" etc. I think their is no need to use such terms. The interest and the quality of the work is obvious for the reader, and the use of such formulations may appear pretentious.

Response 4:

We agree with your comment and have made modifications to the following sentences in the discussion section.

Line 404 “This is the first study to characterize the induction of hybrid immunity in children with cancer” was changed to: “This study characterizes the induction of hybrid immunity in children with cancer”

Line 519 “Overall this study provides considerable insight into the immune response following vaccination” was changed to “Overall this study provides insight into the immune response following vaccination”

Some typing errors were found within the text (not exhaustive):

-line 188 repetition "in in"

-line 374 SARS-COV-2 instead of Sars-CoV-2

Response typing errors:

Thank you for pointing out those typing errors.

In line 227 the extra repetition of “in in” was removed, changing it to “in”.

In line 456 “Sars-COV-2” was changed to “SARS-CoV-2”. 

Reviewer 2 Report

This is a very well-written manuscript describing the importance of Covid-19 vaccination in children under cancer treatment. The authors have explained very well the preference and limitations of this study, which is much appreciated. The study provides significant results in recommending a 3-dose series of BNT162b2 mRNA covid vaccination; children under active cancer treatment should be eligible for a three-dose vaccination regimen to prevent covid-19 infections in them. Overall the study is a good fit, and These are minor comments that can help improve this manuscript.

1.  Authors should provide detailed explanations for both figures 1 & 2 in the text. It is unclear how the authors correlate the clinical data in the tables to the corresponding figures. It needs more explanation in the results section. 

2. Authors should elaborate more on the specific discriminations they have seen in patients with solid tumors vs those with hematological malignancies. Although they have mentioned in the discussion about it yet, it can be further improved by providing essential references on why these differences are observed and what could be done to improve the treatment in children with solid or CNS patients

3. There are a few typo mistakes that should be addressed by the authors

with the changes mentioned above; this manuscript can be improved for the broad audience of Cancers. It will be a great study to support further the importance of booster doses for Covid-19 in patients undergoing cancer therapies. 

Author Response

Response to Reviewer 2

This is a very well-written manuscript describing the importance of Covid-19 vaccination in children under cancer treatment. The authors have explained very well the preference and limitations of this study, which is much appreciated. The study provides significant results in recommending a 3-dose series of BNT162b2 mRNA covid vaccination; children under active cancer treatment should be eligible for a three-dose vaccination regimen to prevent covid-19 infections in them. Overall the study is a good fit, and These are minor comments that can help improve this manuscript.

Response

Thank you for your valuable comments.

We hope to have sufficiently addressed your concerns.

1. Authors should provide detailed explanations for both figures 1 & 2 in the text. It is unclear how the authors correlate the clinical data in the tables to the corresponding figures. It needs more explanation in the results section.

Response 1:

Both figures 1 & 2 show the individual values of the B-cell and T-cell response for each patient. Also the median antibody titers (figure 1A, 1C) and median amount of interferon-gamma (IFN-γ) as released by SARS-CoV-2 specific T-cells (figure 2A) are shown in these figures and also in table 2 and table 3 (lines 259-260 and 280-281 for figure 1A, lines 273-274 for figure 1C and lines 327 – 330 for figure 2A). We specifically mention each figure with the matching, corresponding table in the same sentence to indicate their relatedness.

To improve the visibility of the relation between the figures and the tables, we modified table 2 and table 3. The treatment groups (Tx < 6 weeks, Tx > 6 weeks and No Tx) were switched from the rows to the columns, while the response groups (non-responders, low-responders and good-responders) were switched from the columns to the rows. In this way, the format of the table 2 and table 3 follows the exact same format of the figures.

Furthermore the p-value of the Mann-Whitney U test testing a significant difference between the Tx < 6 weeks group and Tx > 6 weeks group is depicted in figure 1A and figure 2A and also mentioned in the text (line 259 for figure 1A). Since we saw that this wasn’t the case for figure 2A, we also added the result of the Mann-Whitney U test of figure 2A (P = 0.0083) to the text .

As a result lines 327-330 changed from The median (range) amount of IFN-γ release was significantly lower in the Tx < 6 weeks group (841 [0-1890] mIU/mL) compared with the Tx > 6 weeks group (1890 [264-1890] mIU/mL)” to “The median (range) amount of IFN-γ release was significantly lower in the Tx < 6 weeks group compared with the Tx > 6 weeks group (841 [0-1890] mIU/mL vs 1890 [264-1890] mIU/mL; P = 0.0083).

We hope these changes improve the relatedness between the figures and the tables.

2. Authors should elaborate more on the specific discriminations they have seen in patients with solid tumors vs those with hematological malignancies. Although they have mentioned in the discussion about it yet, it can be further improved by providing essential references on why these differences are observed and what could be done to improve the treatment in children with solid or CNS patients

Response 2:

Thank you for your suggestion. It would have been interesting to elaborate more on the specific discriminations between patients with solid tumors and hematological malignancies. Unfortunately the sample sizes of those 2 groups in our manuscript is too small to explore this aspect in depth. We however refer to larger studies in adults that have more power to address these differences, one of which a systematic review (lines 493-507).  

3. There are a few typo mistakes that should be addressed by the authors

Response 3:

Thank you for noticing. We corrected the following typing errors:

In line 59 “2%-7%” was changed to “2-7%”.

In line 187 “Patient” was changed to “Patients”.

In line 197 “CAR-T cell therapy” was changed to “CAR T-cell therapy”.

In line 207 and 274 an extra blank space was added, changing “Tx <6 weeks” to “Tx < 6 weeks”.

In line 227 the extra repetition of “in in” was removed, changing it to “in”.

In line 259 a blank space was removed changing “[0.10 -5091]” to “[0.10-5091]”.

In line 302 and 370 “CAR-T therapy” was changed to “CAR T-cell therapy”.

In line 375 the range was added changing “Median (range) SARS-CoV-2 specific IFN-γ-release in the Tx < 6 weeks group was 669 mIU/mL” to Median (range) SARS-CoV-2 specific IFN-γ-release in the Tx < 6 weeks group was 669 (54-1890) mIU/mL”.

In line 380 “1 dose-vaccination” was changed to “1-dose vaccination”.

In line 383 “responders” was changed to “responder” and “responder” was changed to “responders.”

In line 456 “Sars-COV-2” was changed to “SARS-CoV-2”.

In line 459 “T-cel responders” was changed to “T-cell responders”.

In the whole manuscript (table 1, the legends of figure 1B and 2B) “CAR-T” was changed to “CAR T”.

In the whole manuscript “first”, “second”, “third” was changed to “1st”, “2nd”, “3rd whenever mentioned in a sentence (line 111, 113, 114, 146, 151, 155, 225, 229, table 1, 253, 269, 271, 276, 351, 374, 413, 422, 429).

In table 2 the percent sign (%) was added after the number of good responders in the Tx > 6 weeks group in patients with 3 immunizing events. Also in the footnotes“non responders” was changed to “non-responders”.

In table 3, “Hybrid (2 vaccination + SARS-CoV-2 infection)c was changed to “Hybrid (2 vaccinations + SARS-CoV-2 infection)c. In the footnotes: “non responders” was changed to “non-responders”.

with the changes mentioned above; this manuscript can be improved for the broad audience of Cancers. It will be a great study to support further the importance of booster doses for Covid-19 in patients undergoing cancer therapies.

Response: Thank you.

Reviewer 3 Report

The manuscript, entitled Immune response following Covid-19 vaccination in paediatric cancer patients, demonstrates the vaccine strategy that should be used in the study in the paediatric population and further suggests that the third BNT162b2 mRNA COVID-19 vaccine dose in this population should be considered.

1.     The present study holds limited novelty, since similar study has already been published (PMID: 35763647) recently, and they have already explored the immune response to BNT162b2 mRNA COVID-19 vaccination in Children and Adolescents with Cancer and Hematologic Diseases and concluded that an additional dose for the paediatric cancer patients should be considered.

2.     The microneutralization assay to assess the neutralizing capacity of neutralizing antibodies (nAbs) as the key protective indicators of SARS-CoV-2 is highly recommended.

3.     Please modify the title to indicate the immune response to the BNT162b2 mRNA COVID-19 vaccination since the study aims to evaluate the antibody and T-cell response in children, particularly with BNT162b2 mRNA COVID-19 vaccine, otherwise the title is misleading.

4.     Line 48 – Kindly rephrase adult patients with cancer to adult cancer patients.

5.     Section 2 - Materials and Methods – Please break up the sub-section Study design and participants into distinct sections, such as study population and immunization schedule, as this would help the reader understand the subject with clarity.

6.     Kindly replace IFN-y with IFN-γ in the text and figure 2 legend respectively.

7.     Kindly rearrange the key/legend of the Figure 2A properly.

8.     Line 403: Kindly replace 200/mm3 with 200/mm3.

9.     Kindly discuss strength of the study in the discussion section.

10.  I would like the authors to discuss the factor of waning vaccine immunogenicity as a significant issue, and the need for a 3rd dose of the vaccine should be considered.

11.  Finally, this article matches the scope of vaccine-related research, and not directly fulfilling the scope of this journal, as the study does not directly contribute to further the fundamental component of cancer research.

I'm not sure if this study is a good fit for this journal.

Author Response

Response to Reviewer 3

The manuscript, entitled Immune response following Covid-19 vaccination in paediatric cancer patients”, demonstrates the vaccine strategy that should be used in the study in the paediatric population and further suggests that the third BNT162b2 mRNA COVID-19 vaccine dose in this population should be considered.

Thank you for the insightful and helpful comments. We hope that we have sufficiently addressed all comments.

1. The present study holds limited novelty, since similar study has already been published (PMID: 35763647) recently, and they have already explored the immune response to BNT162b2 mRNA COVID-19 vaccination in Children and Adolescents with Cancer and Hematologic Diseases and concluded that an additional dose for the paediatric cancer patients should be considered.

Response 1:

Thank you for your comment. We do agree that this is not the first paper on the immune response following COVID-19 vaccination in pediatric cancer patients. However we do think the findings from this study are valuable and unique due to its larger sample size and because of the inclusion of children aged 5-11 years and the inclusion of children that have a history of SARS-CoV-2 infection.

2. The microneutralization assay to assess the neutralizing capacity of neutralizing antibodies (nAbs) as the key protective indicators of SARS-CoV-2 is highly recommended.

Response 2:

Thank you for this suggestion. We didn’t have the opportunity to also do a micro neutralization assay. However, we believe the multiplex fluorescent bead-based immune assay we used to measure the antibody concentrations is a valid method that has also proven to yield robust results (PMID 32766833, PMID: 35114690, PMID: 34767759, PMID: 36354032 amongst others).

3. Please modify the title to indicate the immune response to the BNT162b2 mRNA COVID-19 vaccination since the study aims to evaluate the antibody and T-cell response in children, particularly with BNT162b2 mRNA COVID-19 vaccine, otherwise the title is misleading.

Response 3:

We agree with your comment and have made some modifications to the title to make it clear that patients were vaccinated with the BNT162b2 mRNA COVID-19 vaccine. We also changed “Covid-19 vaccination” to “COVID-19 vaccination”. The previous title was: “Immune response following Covid-19 vaccination in pediatric cancer patients” and the new title now reads: “Immune response following BNT162b2 mRNA COVID-19 vaccination in pediatric cancer patients”.

4. Line 48 – Kindly rephrase adult patients with cancer to adult cancer patients.

Response 4:

Thank you for your suggestion. We rephrased the sentence on line 49 “Earlier research found adult patients with cancer to” to “Earlier research found adult cancer patients to”.

5. Section 2 - Materials and Methods – Please break up the sub-section Study design and participants into distinct sections, such as study population and immunization schedule, as this would help the reader understand the subject with clarity.

Response 5:

Thank you for the suggestion. To increase the readability of the subsection “2.1 Study design and participants” we moved the part about the immunization- and sampling schedule:

“To assess the antibody- and T-cell response, blood was sampled 28 (21-42) days after the second vaccination. For immunocompromised children aged 12 and above, at a later stage a 3rd vaccination dose was included in the national vaccination program. Patients (12-17 years) already included in the study, who received a third vaccination, were asked for permission for an extra blood sample 28 (21-42) days after the third vaccination.”

to subsection “2.2 Procedures/ blood sample collection and laboratory analyses” (line 110-114).

6. Kindly replace IFN-y with IFN-γ in the text and figure 2 legend respectively.

Response 6:

We corrected “IFN-y” too “IFN-γ” throughout the manuscript (lines 130, 132, 205, 328, 361, 375, 384) & the Y-axes of figure 2A and figure 2B).

7. Kindly rearrange the key/legend of the Figure 2A properly.

Response 7:

Thank you for noticing. We rearranged the legend for figure 2A. The legend has been modified by adding a border to it, conform the legends in the other figures. Also the legend has been moved to the right so that it aligns with the other text in figure 2A. 

8. Line 403: Kindly replace 200/mm3 with 200/mm3.

Response 8:

Thank you for pointing this out. We have changed in line 493 “200/mm3” to “200/mm3”.

9. Kindly discuss strength of the study in the discussion section.

Response 9:

To better indicate the strengths of the study, we have added the following sentences to the discussion section (lines 510-517)

“Despite these limitations, the heterogeneity of the included patients is also one of the strengths of this study. Both the studies from Lehrnbecher et al and Poparn et al reported data from children older than the age of 12 and all without a history of SARS-CoV-2 infection [26,27]. Data on the immune response following COVID-19 vaccination is therefore scarce in children aged 5-11 years and in children with a hybrid immunity. As this study did include children with these characteristics, the current results contribute to knowledge on COVID-19 vaccination during and after cancer treatment and also specifically in younger children and in previously infected children.”

10. I would like the authors to discuss the factor of waning vaccine immunogenicity as a significant issue, and the need for a 3rd dose of the vaccine should be considered.

Response 10:

Thank you for your suggestion. In the discussion section, we describe our plans for future research on the long-term humoral and cellular immunity, and on clinical efficacy, 1 year after vaccination (line 467-469). We also referred to studies that reported on the importance of the cellular immune response as the humoral immune response was proven to decrease more quickly over time (line 470 – 476).  

As we agree on the significance of waning immunity, we have added a few sentences to this part to further elaborate on this topic.

Lines 476-484

“Several studies reported a waning humoral immune response following COVID-19 vaccination [32,33] or SARS-CoV-2 infection [34]. A substantial decrease in IgG levels and neutralizing antibody titers was seen in the 6 months following BNT162b2 COVID-19 vaccination in adults [32]. This waning of immunity, already typically starts 3-24 weeks after full vaccination [33]. Waning is even more in immunocompromised patients and is also seen after a 3rd vaccination [35]. Also in children with a history of SARS-CoV-2 infection waning of humoral immune response was seen in the 14 months following infection [34]. Therefore future research is needed to optimize the booster schedule in children with cancer [36]”.   

11. Finally, this article matches the scope of vaccine-related research, and not directly fulfilling the scope of this journal, as the study does not directly contribute to further the fundamental component of cancer research.

I'm not sure if this study is a good fit for this journal.

Response 11:

Thank you for your remark. We think this study adds to the limited knowledge on COVID-19 vaccination in pediatric cancer patients and to the knowledge on vaccination in general during and after intensive cancer treatment. We hope it can make a valuable contribution to  future vaccination strategies in those patients thereby improving the supportive care for children with cancer.

Round 2

Reviewer 3 Report

Thank you for addressing the raised concerns.